# Preparation of Hybrid Nanopigments with Excellent Environmental Stability, Antibacterial and Antioxidant Properties Based on *Monascus* Red and Sepiolite by One-Step Grinding Process

**DOI:** 10.3390/nano13111792

**Published:** 2023-06-02

**Authors:** Shue Li, Penji Yan, Bin Mu, Yuru Kang, Aiqin Wang

**Affiliations:** 1Key Laboratory of Clay Mineral Applied Research of Gansu Province, Center of Eco-Material and Green Chemistry, Lanzhou Institute of Chemical Physics, Chinese Academy of Sciences, Lanzhou 730000, China; seli17@licp.cas.cn (S.L.); yurukang@licp.cas.cn (Y.K.); 2Center of Materials Science and Optoelectronics Engineering, University of Chinese Academy of Sciences, Beijing 100049, China; 3College of Chemistry and Chemical Engineering, Key Laboratory of Hexi Corridor Resources Utilization of Gansu Province, Hexi University, Zhangye 734000, China; yanpenji@hxu.edu.cn

**Keywords:** *Monascus* red pigment, sepiolite, hybrid nanopigments, environmental stability, superamphiphobic coatings

## Abstract

This study is focused on the preparation, characterization, and multifunctional properties of intelligent hybrid nanopigments. The hybrid nanopigments with excellent environmental stability and antibacterial and antioxidant properties were fabricated based on natural *Monascus* red, surfactant, and sepiolite via a facile one-step grinding process. The density functional theory calculations demonstrated that the surfactants loaded on sepiolite were in favor of enhancing the electrostatic, coordination, and hydrogen bonding interactions between *Monascus* red and sepiolite. Thus, the obtained hybrid nanopigments exhibited excellent antibacterial and antioxidant properties, with an inhibition effect on Gram-positive bacteria that was superior to that of Gram-negative bacteria. In addition, the scavenging activity on DPPH and hydroxyl free radicals as well as the reducing power of hybrid nanopigments were higher than those of hybrid nanopigments prepared without the addition of the surfactant. Inspired by nature, gas-sensitive reversible alochroic superamphiphobic coatings with excellent thermal and chemical stability were successfully designed by combining hybrid nanopigments and fluorinated polysiloxane. Therefore, intelligent multifunctional hybrid nanopigments have great application foreground in related fields.

## 1. Introduction

Natural colorants are generally extracted from or produced by natural sources, including animals, plants, and microorganisms [1]. *Monascus* pigments, a large category of secondary pigments with a similar azaphilone skeleton produced by the fungus Monascus, have served as natural food colorants and traditional Chinese medicines in East Asian countries for thousands of years [2]. Like other natural pigments, *Monascus* pigments also have various biological activities, bright colors, no obvious side effects, and high safety [3,4,5]. In contrast to the inhibition of amination under acidic conditions, an alkaline environment has a significant promoting effect on the amination of orange *Monascus* pigments, resulting in the formation of *Monascus* red (MR) with amination [3]. As the main components of MR, the two red pigments are very similar in structure, except for the different side chain lengths. Compared with beetroot red and paprika extract, MR has a more vivid red hue and stronger coloring power. In addition, the insoluble MR has been transformed into a water-soluble pigment in the extraction process to broaden the range of applications. Although MR presents good solubility, the shortcomings in the thermostability, antibacterial, and antioxidant properties of naturally occurring pigments severely hinder their practical applications, such as color fading after being exposed to external environments [6,7,8].

Maya blue, a well-known hybrid nanopigment, has bright colors and extraordinary stability, the main components of which were identified as attapulgite (APT) and the natural pigment indigo [9,10]. Inspired by Maya blue, APT and sepiolite (Sep) with similar crystal textures have been widely exploited to encapsulate natural and synthetic organic dyes for preparing Maya blue-like hybrid nanopigments. Compared with APT, Sep is a magnesium hydrosilicate with larger nanochannels (1.06 nm × 0.37 nm) and fibrous morphology, which is also formed by two continuous tetrahedral sheets of silicon oxides linked by discontinuous octahedral sheets of magnesium oxides [11]. The reversal period of the apical oxygen in every three silicate chairs results in the existence of numerous silanol (Si–OH) groups on the surface and the formation of parallel tunnels and channels arranged along the fibers [12,13]. The unique porous structure is filled with weakly bound zeolitic H_2_O that forms the hydrogen bridge with the tetrahedral sheet and with tightly bound structural OH_2_ coordinated to the magnesium at the ribbon borders [14,15]. The discontinuity of the external silica sheet allows the penetration of natural pigments into the nanochannels of Sep, providing a protective effect, and the larger nanochannels of Sep may accommodate organic dyes with large sizes, which cannot enter the nanochannels of APT (0.64 nm × 0.37 nm). For example, Szadkowski et al. fabricated high-performance hybrid nanopigments based on anthraquinone chromophores and clay minerals (APT and Sep) [16]. The obtained hybrid pigments had improved chemical and thermal resistance and acid/base allochroic behavior. In addition, Silva et al. designed a highly fluorescent hybrid nanopigment based on Sep and flavylium cations that exhibited excellent stability against thermal degradation and aqueous solution [17]. However, it is scarcely focused on the functionalization of hybrid nanopigments based on natural pigments and clay minerals, except for environmental stability. It was reported that the surface modification of clay minerals could enhance the surface reactivity and the adsorption capacity for organic molecules [18,19], and the incorporation of a few surfactants also provided antibacterial activity for clay minerals [20].

Herein, the present focus is on the design of multifunctional hybrid nanopigments composed of MR and Sep in the presence of cetyldodecyltrimethyl ammonium bromide (DTAB) or cetyltrimethyl ammonium bromide (CTAB) by combining grinding and heating treatment. The aim of the present work is to obtain intelligent multifunctional hybrid nanopigments to improve the environmental stability of natural MR molecules. A combination of systematic characterization and density functional theory (DFT) calculations was applied to reveal the possible interactions among MR, Sep, and surfactant molecules. The environmental stability, antibacterial, and antioxidant properties of the as-prepared hybrid nanopigments were investigated in detail. Furthermore, the obtained hybrid nanopigments were employed to prepare the colorful superamphiphobic coatings by modification with fluorinated polysiloxane, and the reversible acid/base allochroic properties of hybrid nanopigments and colorful superamphiphobic coatings were also explored. The multifunctional hybrid nanopigments are expected to be applied in relevant fields in the future.

## 2. Materials and Methods

### 2.1. Materials

*Monascus* red (MR) was supplied by Maya Reagent Co., Ltd. (Jiaxing, China). Sepiolite (Sep) was obtained from Yixian Dazhi Insulation Materials Sepiolite Co., Ltd. (Baoding, China) and was treated using 4% hydrochloric acid to remove the associated carbonates and silica sand. The main chemical compositions of Sep consisted of CaO (7.87%), Al_2_O_3_ (4.76%), MgO (25.66%), SiO_2_ (49.35%), K_2_O (0.93%), Na_2_O (4.93%), and Fe_2_O_3_ (1.39%), as determined by X-ray fluorescence (XRF). Dodecyltrimethyl ammonium bromide (DTAB) was obtained from Tianjin Kemiou Chemical Reagent Development Center (Tianjin, China). Cetyltrimethyl ammonium bromide (CTAB) and TEOS (99.9%) were provided by Chengdu Cologne Chemicals Co., Ltd. (Chengdu, China). PFDTES (97%) were obtained from Gelest. The substrates were glass slides (24 × 50 mm, Braunschweig, Germany).

### 2.2. Preparation of Hybrid Nanopigments

The hybrid nanopigments were prepared by a simple one-step grinding method followed by heat treatment. First, 0.25 g of MR, 0.0625 g of DTAB, 1.00 mL of deionized water, and 1.00 g of Sep were added to a mortar and ground for 30 min, followed by heat treatment at 120 °C for 4 h in an oven. Subsequently, the sample was washed with 50.0 mL of deionized water three times, and the solid product was dried at 60 °C and marked as DTAB/MR/Sep. The preparation process for CTAB/MR/Sep was the same as that of DTAB/MR/Sep except for replacing DTAB with CTAB. The as-prepared hybrid nanopigments without the addition of surfactants were labeled MR/Sep. The computational details and models of hybrid nanopigments are described in the Appendix A.

### 2.3. Preparation of Hybrid Nanopigments@fluoro POS Coatings

The DTAB/MR/Sep powder (0.10 g) was dispersed into a mixture of EtOH (8.80 mL) and ammonia aqueous solution (1.20 mL). A certain amount of TEOS and PFDTES were added to the above mixture after ultrasonic treatment for 30 min. Then, the DTAB/MR/Sep@fluoro POS suspension was formed by magnetic stirring at 600 rpm for 6 h. The DTAB/MR/Sep@fluoro POS coatings were obtained by spray-coating the DTAB/MR/Sep@fluoro POS suspension onto glass slides using an airbrush (INFINITY 2 in 1, Harder & Steenbeck, Germany) with 0.2 MPa N_2_ at a distance of 15 cm.

### 2.4. Thermal Stability of Hybrid Nanopigments

To evaluate the thermal resistance of the synthesized hybrid nanopigments, MR, MR/Sep, DTAB/MR/Sep, and CTAB/MR/Sep were in turn placed in an oven at 120, 150, and 180 °C for 30 min. The colorimetric values and colors of the samples before and after thermal treatment at different temperatures were recorded to evaluate their thermal stability.

### 2.5. DPPH Scavenging Activity

The DPPH (1,1-diphenyl-2-picrylhydrazyl) reactivity of the materials was assayed with a previous method with some slight modifications [21]. In brief, 1.00 mL of individual test sample solution (5.00 mg/mL, dissolved in DMSO) was added to 4.00 mL of DPPH solution (0.10 mM, dissolved in EtOH) and stood for 30 min in complete darkness. And then the absorbance of all samples was determined. The formula for DPPH scavenging activity was calculated as follows (Equation (1)):DPPH scavenging activity (%) = [*A_c_* − (*A_s_* − *A_b_*)]/*A_c_* × 100%(1)
where, *A_c_* and *A_s_* are the absorbance values at 517 nm of the control and the test sample, respectively. *A_b_* is the absorbance of the test sample without DPPH solution.

### 2.6. Hydroxyl Radical Scavenging Activity

The antioxidant activity of the samples against hydroxyl radicals was measured depending on the method described by Zhang et al. [22]. Two milliliters of sample solution (5.00 mg/mL, dissolved in DMSO) was mixed with FeSO_4_ solution (2.00 mL, 9.00 mM) and salicylic acid EtOH solution (2.00 mL, 9.00 mM). The reaction was launched by the introduction of 2.00 mL of H_2_O_2_ (8.80 mM) to the above mixture and performed at 37 °C for 30 min under dark conditions. The capability of scavenging hydroxyl radicals was calculated according to (Equation (2)):Hydroxyl radical scavenging activity (%) = [*A_c_* − (*A_s_* − *A_b_*)]/*A_c_* × 100%(2)
where, *A_c_* and *A_s_* are the absorbances at 510 nm of the control and the test sample, respectively. *A_b_* is the absorbance of the sample without H_2_O_2_.

### 2.7. Reducing Power

The reducing power of the test samples was measured using a previous method with some slight modifications [21]. Briefly, 1 mL of each sample solution (5.00 mg/mL, dissolved in DMSO) was added to 2.50 mL of K_3_Fe(CN)_6_ solution (1%, *w*/*v*) and 2.50 mL of PBS (0.20 M, pH 6.6). The above mixture stood for 20 min in the water bath at 50 °C, followed by the addition of 2.50 mL of trichloroacetic acid (10%, *w*/*v*) and centrifugation (4000 rpm, 10 min). Then, 2.50 mL of the above-mentioned supernatant was mixed with deionized water (2.50 mL), freshly prepared 0.1% *w*/*v* FeCl_3_ solution (0.50 mL), and put aside for 10 min under greenhouse conditions. Finally, the absorbance of all samples at 700 nm was documented.

### 2.8. Antibacterial Activity of Hybrid Nanopigments

Minimum inhibitory concentrations (MICs) of MR, MR/Sep, DTAB/MR/Sep, and CTAB/MR/Sep against five kinds of bacteria, including *Staphylococcus aureus* (*S. aureus*), *methicillin-resistant Staphylococcus aureus* (*MRSA*), *Enterococcus faecalis* (*E. faecalis*), *Escherichia coli* (*E. coli*), and *Pseudomonas aeruginosa* (*P. aeruginosa*) were measured by the twofold dilution method. Typically, an appropriate amount of MR, MR/Sep, DTAB/MR/Sep, and CTAB/MR/Sep was put in Petri dishes, and then they were disinfected under the UV lamp for 30 min. 20 mL of NA were added to the Petri dish with sterilized samples at different concentrations after being autoclaved and cooled to 50–60 °C. Next, 1 μL of bacterial solution (104 CFU/mL) was inoculated in a Petri dish. Three spots were inoculated in each Petri dish. In addition, a positive control with no test sample was also established. All Petri dishes were incubated in a carbon dioxide incubator for 16–20 h at 35 °C. The MIC value of the obtained sample was the minimum sample concentration, which completely inhibited colony growth. The MIC value of the test sample was the minimum concentration, which completely inhibited colony growth.

### 2.9. Characterization Methods

The Fourier Transform Infrared Spectrometer (FTIR) spectra were carried out on an FTIR spectrometer (Frontier, PerkinElmer, Waltham, MA, USA) in the range of 4000~400 cm^−1^ using KBr pellets. X-ray diffractometer (XRD) patterns were taken using a SmartLab SE multifunctional X-ray diffractometer (Rigaku Co., Tokyo, Japan) in the range of 2*θ* = 3°~80°. The surface morphology of the samples was observed on a JEM-2100 transmission electron microscope (TEM, JEOL, Ltd., Tokyo, Japan). X-ray photoelectron spectroscopy (XPS) was recorded on an ESCALAB 250Xi electronic spectrometer with 300 W Al Kα radiation (Thermo Fisher Scientific, Waltham, MA, USA). Thermal gravimetric analysis (TGA) was measured with a simultaneous thermal analyzer (STA 8000, PerkinElmer, Waltham, MA, USA) in the temperature range from 30 to 800 °C with a heating rate of 10 °C/min under N_2_ atmosphere. The color parameters of the samples were recorded on a ColorEye automatic differential colorimeter (X-Rite, Ci 7800) according to the 1976 *L***a***b** colorimetric method of the Commission International de l’Eclairage (CIE). The contact angle (CA) and low sliding angle (SA) were recorded by the Contact Angle System OCA 20, Dataphysics (Baden-Wuerttemberg, Germany), equipped with a tilting table. The syringe was located such that the water droplet (7 mL) contacted the sample surface before leaving the needle. The reported data were the average of three results at different locations on the samples.

## 3. Results

### 3.1. Synthesis and Characterization of Hybrid Nanopigments

The hybrid nanopigments were prepared by a one-step grinding process combining MR, Sep, and surfactant. MR with large molecular size was difficult to enter into the internal channels of Sep. Therefore, the binding of MR molecules to Sep was first promoted by means of grinding. After that, organic modifiers were incorporated and mainly adsorbed on the surface and/or loaded at the edge of Sep to enhance the interaction between natural pigments and clay minerals [23]. In this way, the red hybrid pigments were successfully obtained.

To ascertain the phase change during the preparation of the as-synthesized samples, XRD patterns of the obtained powders are illustrated in Figure 1a. The diffraction peaks of Sep at 2*θ* = 7.37°,12.00° 19.45°, and 24.38° were displayed in the XRD pattern of Sep (JCPDS PDF card No. 75-1597) [24], which corresponded to *d_110_* of 12.01 Å, *d_130_* of 7.25 Å, *d_031_* of 4.56 Å, and *d_151_* of 3.65 Å according to Bragg’s law, respectively. In addition, two reflections at 2*θ* = 9.50° and 28.64° correspond to talc, while the reflections appearing at 2*θ* = 30.99°, 41.17°, 44.97°, and 50.55° were attributed to dolomite. According to previous works, the organic modifiers did not change the *d_110_* spacing due to the adsorption on the Sep surface and/or loading at the edge of Sep [25]. The *d_110_* spacing of DTAB/MR/Sep (12.12 Å) and CTAB/MR/Sep (12.05 Å) was similar to that of Sep (11.99 Å), and thus MR molecules might be adsorbed on the surface of Sep. Compared with Sep, the reflection plane (110) of hybrid nanopigments presented a lower relative intensity, which was related to the decrease in the Sep stacking order after the incorporation of organic modifiers and MR with the assistance of mechanical grinding [26]. Moreover, the XRD patterns of the above samples had the same characteristic reflections as Sep, indicating that the structure of Sep was not damaged in the presence of organic modifiers and MR during the grinding process. Meanwhile, no reflections corresponding to MR were displayed in the XRD patterns of as-prepared samples, indicating that MR was rearranged in an amorphous state after being interacted with Sep.

The morphologies of Sep and MR/Sep were observed using TEM, and the images are presented in Figure 1b,c. It could be observed that Sep with typical fibrous morphology exhibited massive agglomeration due to the existence of van der Waals forces and hydrogen bonding among them (Figure 1b). These nanofibers connected to each other resulted in poor dispersion of Sep. In the case of MR/Sep and DTAB/MR/Sep hybrid nanopigments, the Sep nanofibers with a diameter of 20–60 nm were evidently dissociated rather than tightly packed together due to the grinding process and modification (Figure 1c and Appendix A).

Pore structural parameters and electrostatic interactions are also major properties in the formation of hybrid nanopigments based on surfactants, MR and Sep. The modifications of the samples had no effect on the structures compared with the raw Sep since the modifications mainly occurred on the Sep surface, keeping the structure intact. Variations in the specific surface areas (*S_BET_*) and the total pore volume (*V_total_*) of Sep modified with two surfactants can be observed in Table 1. Compared with Sep, the *S_BET_* of DTAB/Sep and CTAB/Sep decreased from 31.73 m^2^/g to 30.57 m^2^/g and 17.01 m^2^/g, respectively. It might be due to the fact that organic molecules were mainly attached to the Sep surface, causing a decrease in *S_BET_* [25]. An increase in *V_total_* suggested that the modified Sep with surfactants could serve as an excellent candidate host for loading MR. Furthermore, a significant change was obtained in the *S_BET_* and *V_total_* values of DTAB/MR/Sep and CTAB/MR/Sep, and the decrease in the *S_BET_* and *V_total_* of hybrid nanopigments clearly indicated the successful incorporation of the MR molecule on the surface of Sep. As shown in Table 1, the negatively charged MR exhibited zeta potentials of −35.25 mV. The isomorphous substitution resulted in negative charges of Sep (−12.63 mV). By contrast, dodecyltrimethylammonium cation (DTAB^+^) and cetyltrimethylammonium cation (CTAB^+^) were positively charged as the typical cationic surfactants. Therefore, the negative zeta potentials of DTAB/Sep (−12.43) and CTAB/Sep (−2.05) increased slightly due to the formation of the electrostatic interaction between the surfactants and Sep, and then the zeta potential values of DTAB/MR/Sep and CTAB/MR/Sep decreased with the incorporation of the negatively charged MR.

To reveal the chemical changes of samples before and after treatment, the FTIR spectra of MR, Sep, MR/Sep, DTAB/Sep, CTAB/Sep, DTAB/MR/Sep, and CTAB/MR/Sep are presented in Figure 2. The signal at 3374 cm^−1^ confirmed the presence of O–H stretching and bending vibrations of H-bonds [27,28]. The band at approximately 2928 cm^−1^ was due to the C–H stretching mode of the -CH_3_ group [28]. The band at 1631 cm^−1^ corresponded to the stretching mode of C=C on the benzene of the O–H deformation vibration [27]. The peaks at 1542 cm^−1^ and 1463 cm^−1^ were due to the C=C stretching mode of aliphatic side chains and C–H vibration, respectively. The typical absorption peaks at 1151 cm^−1^ and 1080 cm^−1^ were due to the stretching vibration of C–O and angle vibration of O–H, respectively.

In the case of Sep (Figure 2a), the O–H stretching vibrations of Mg-bonded hydroxyl groups, coordinated water, zeolitic water, and adsorbed water were observed at 3677 cm^−1^, 3570 cm^−1^, and 3425 cm^−1^, respectively [29]. The band appearing at 1640 cm^−1^ corresponded to the hydroxyl bending vibration of zeolitic water molecules. The 1018~465 cm^−1^ region presented typical absorption bands correlated with the vibrations of Si–O bonds, Si-O-Si groups in the silicate structure, and Si–O–Mg groups [30,31]. As shown in Figure 2b–d, the spectra for DTAB/Sep and CTAB/Sep exhibited new characteristic bands between 2929 cm^−1^ and 2854 cm^−1^ and a region between 1467 and 1440 cm^−1^. The former was due to νas(C-H) and νs(CH-), respectively, while the latter was attributed to the bending vibrations of [N(CH_3_)_3_] and (-CH_2_) groups [32]. It indicated that DTAB and CTAB were successfully loaded on Sep. For the MR/Sep, DTAB/MR/Sep, and CTAB/MR/Sep hybrid nanopigments, the typical absorption bands at approximately 1547~1542 cm^−1^ and 1462 cm^−1^ were due to the presence of MR molecules. On the other hand, the O–H groups of Sep located at 3425 cm^−1^ and 1640 cm^−1^ were shifted after adding MR and organic modification, respectively. It could be explained by the fact that there was hydrogen bond interaction between the surfactants, MR and Sep.

The surface chemical compositions of DTAB/Sep, MR/Sep, and DTAB/MR/Sep were studied by XPS spectra. As illustrated in Appendix A, DTAB/Sep, MR/Sep, and DTAB/MR/Sep were mainly composed of carbon, oxygen, silicon, and magnesium. Compared with DTAB/Sep and MR/Sep, the carbon content of DTAB/MR/Sep increased from 35.78% and 36.36% to 52.35%, while the content of oxygen, silicon, and magnesium elements decreased (Appendix A). The binding energy of Si2p and Mg1s electrons in DTAB/MR/Sep was lower than in DTAB/Sep and MR/Sep. This was probably due to the polarization of the Si–O bond and the Mg–O bond during the interaction of DTAB, MR, and Sep [33]. As displayed in Appendix A, the fitting peaks were ascribed to C–H/C–C and C=O bonds in the high-resolution C1s spectra of DTAB/Sep, MR/Sep, and DTAB/MR/Sep, respectively.

The TGA and DTG curves were applied to explore the thermal stability of the as-prepared samples (Figure 3). The weight loss of Sep before 100 °C was related to the removal of physically adsorbed H_2_O and zeolitic water [34]. The first maximum degradation temperature of Sep occurred before 100 °C. The final maximum degradation temperature of Sep was located between 600 and 700 °C. The remaining mass loss started at approximately 550 °C, which was put down to the dehydroxylation of the structural OH associated with the octahedral sheet [11]. For the organic-modified Sep, the mass loss in the temperature range between 200 °C and 450 °C was ascribed to the decomposition of organic modifiers [35]. It suggested that the surfactant molecules were tightly bonded to the surface and the edge of Sep, leading to an increase in their degradation temperature. The final maximum degradation temperature of organic-modified Sep was enhanced compared with Sep. As shown in Appendix A, the first stage from 30 to 180 °C corresponded to water evaporation [36], and the mass loss of 180–600 °C was attributed to the decomposition of MR. Further mass loss occurred at 650–800 °C and corresponded to the carbonization of MR under the nitrogen atmosphere. For hybrid nanopigments, the TGA and DTG curves roughly resembled those of Sep and the corresponding modified Sep before 180 °C, and the mass loss of hybrid nanopigments was attributed to the adsorbed water. The decomposition of MR began at approximately 200 °C, which proved that the introduction of Sep and surfactants could improve the thermal stability of MR molecules. Depending on TGA results, it could be calculated that the mass loss of MR in hybrid nanopigments was 2.82%, 7.72%, and 4.88% for MR/Sep, DTAB/MR/Sep, and CTAB/MR/Sep, respectively. It showed that the addition of surfactants was also in favor of increasing the loading of MR on Sep.

### 3.2. The Possible Interaction between DTAB, MRP and Sep

The model structures of Sep and MR were designed and optimized at M062X(D3)/6-31G(D)/SMD level (Figure 4). As shown in Figure 4a, the O–Mg bond length of coordination water of Sep was 2.14 Å. The ESP values of MR showed that the minimum value (−114.28 kcal/mol) was at the carbonyl group. The RESP atomic charge also showed that the carbonyl oxygen atom had a smaller negative charge of −0.669, indicating that MR might coordinate with Mg^2+^ in Sep channels or form hydrogen bonds with coordination water. Therefore, MR/Sep-I and MR/Sep-II models were designed to investigate the coordination between the carbonyl group and Mg^2+^ at different positions, respectively. As presented in Figure 4b, the ΔG values of MR/Sep-I and MR/Sep-II in aqueous solution were 7.25 and 6.03 kcal/mol, respectively, while their corresponding O–Mg bond lengths were 2.17 and 2.14 Å, respectively. At the same time, the adsorption structures of MR/Sep-Ⅲ and MR/Sep-Ⅳ were also investigated. However, thermodynamic data displayed that the exothermic values of these two structures were only 0.21 and 0.29 kcal/mol, respectively. In addition, the interaction between DTAB^+^, DTAB, I^−^, and MR was studied. The result indicated that DTAB was adsorbed on the surface and/or loaded at the edge of Sep and then interacted with MR. It could enhance the adsorption capacity of MR molecules in terms of thermodynamics, which was consistent with the experimental results.

In order to further explain the change in free energy, PSI4 was applied to decompose the energy of the composite structure. As listed in Appendix A, the interactions between coordination water and Mg^2+^ in the Sep structure included electrostatic interaction, mutual exclusion, induction, and dispersion, contributing −42.97 kcal/mol, 35.00 kcal/mol, −11.44 kcal/mol, and −8.37 kcal/mol, respectively, with a total energy of −27.78 kcal/mol. Obviously, this value was less than the contribution of electrostatic interaction, induction, and dispersion in MR/Sep-I and MR/Sep-II. It verified that Sep and MR could form relatively stable composite materials through coordination bonds. However, the hydrogen bonds between Sep and MR have weaker interactions than coordination bonds, resulting in less heat release.

### 3.3. Color and Environmental Stability of Hybrid Nanopigments

The color of MR is an important characteristic; that is, bright red is more conducive to dyeing. It appeared that the addition of the appropriate inorganic host played a role in improving the color values of MR. Chromatic CIE coordinates confirmed that the pure MR and hybrid nanopigments were located in the orange and yellowish-pink areas, respectively (Figure 5a). Compared with MR, the *L**, *a**, and *b** values of the series of synthetic samples collected significantly increased. For MR/Sep, DTAB/MR/Sep, and CTAB/MR/Sep, the values of the *L**, *a***,* and *b** parameters were above 8.78, 11, and 11.88, respectively, indicating their colors were bright, red, and yellow, respectively (Appendix A). Similar trends were also observed for the *C** values, which suggested that hybrid pigments presented a more vivid color. The *h^°^* values of the samples were positioned within the quadrants of the CIE *L***a***b** color chart. The differences in *h^°^* demonstrated a clear and comprehensive visual color change, which could also be confirmed by digital photos of the pure MR and hybrid nanopigments. Furthermore, the digital photos and CIE parameters of the pure MR and hybrid nanopigments after being treated at different temperatures are shown in Figure 5b–e. Compared with pure MR, MR/Sep, DTAB/MR/Sep, and CTAB/MR/Sep showed a smaller color difference after heat treatment at different temperatures due to the protection of Sep and the organic-modified Sep matrix.

### 3.4. Antioxidant Activity of Hybrid Nanopigments

The antioxidant activity of hybrid nanopigments was investigated by the determination of DPPH and hydroxyl radical scavenging activity, as well as reducing power. It has been reported that MR has certain antioxidant activities [37]. During the study, the DPPH assay showed 13.78% of radical scavenging activity for MR/Sep, and DTAB/MR/Sep (35.46%), and CTAB/MR/Sep (37.72%) presented higher antioxidant potential than MR/Sep (Figure 6a). Among the free radicals, hydroxyl radicals caused the most damage due to their many active chemical properties and high reaction rate. Therefore, it was often applied to investigate the scavenging capacity of samples on free radicals. Similarly, it indicated that the scavenging effects of MR/Sep on hydroxyl free radicals were weaker than those of the hybrid nanopigments with surfactants. Reducing power is another key indicator to show the potential antioxidant activity of hybrid nanopigments. As presented in Figure 6b, the reducing power of three well-prepared hybrid nanopigments with surfactants increased compared with MR/Sep. In summary, the antioxidant activity of hybrid nanopigments based on MR, Sep, and surfactants was relevant to the amount of MR. The greater the MR amount, the higher the scavenging of free radicals and inhibition of oxidation reactions.

### 3.5. Antioxidant Activity of Hybrid Nanopigments

To evaluate the antibacterial efficiency, Gram-positive bacterial species (*S. aureus*, *MRSA*, and *E. faecalis strains*) and Gram-negative bacterial species (*E. coli* and *P. aeruginosa*) were selected and incubated with five kinds of MR, MR/Sep, DTAB/MR/Sep, and CTAB/MR/Sep samples, respectively. The antibacterial activity of the above sample for five strains was assessed by the MIC values, and the results are shown in Figure 7 and Figure Appendix A, while the MIC values determined for hybrid nanopigments against the above bacterial strains are listed in Appendix A. Pure MR and MR/Sep, as expected, did not inhibit cell growth, whereas DTAB/MR/Sep and CTAB/MR/Sep hybrid nanopigments remarkably inhibited bacterial growth, including *S. aureus*, *MRSA*, and *E. faecalis strains*. The results showed that the bactericidal pattern of the synthesized hybrid nanopigments against *S. aureus*, *MRSA*, and *E. faecalis* strains was again CTAB/MR/Sep > DTAB/MR/Sep. Among them, the MIC values of CTAB/MR/Sep against *S. aureus*, *MRSA*, and *E. faecalis* were 0.10 mg/mL. In this situation, CTAB had the longest alkyl chain length, which was more conducive to cell lysis due to hydrophobic interactions [23].

In contrast, only the DTAB/MR/Sep hybrid nanopigments exhibited bactericidal activity against *E. coli* and *P. aeruginosa*. However, neither cell type was affected by CTAB/MR/Sep to that extent. In the case of DTAB/MR/Sep, it was observed that the Gram-positive bacterial species had a smaller MIC than the Gram-negative bacterial species. This implied that Gram-negative bacterial strains might have higher resistance than Gram-positive bacterial strains. The differences in antibacterial activity could be attributed to the differences in the structure and constituents of the selected bacterial cell membrane. Gram-negative bacteria had been reported to have an outer liposaccharide membrane over a thin peptidoglycan layer, which formed a strong permeability barrier, resulting in *E. coli* and *P. aeruginosa* exhibiting resistance to most available drugs [38]. The antibacterial efficiency of DTAB/MR/Sep and CTAB/MR/Sep was attributed to cell membrane damage due to the interaction between hybrid nanopigments and the bacterial cell wall, except for electrostatic attraction [39].

### 3.6. Application for Reversible Allochroic Superamphiphobic Coatings

Colorful superamphiphobic coatings were obtained by a facile spray method using DTAB/MR/Sep as an example. As shown in Figure 8a, the surface of the DTAB/MR/Sep@fluoro POS nanorods was wrapped with a layer of fluoro POS after modification, and fluoro POS as a cross-linker linked the DTAB/MR/Sep nanorods together, resulting in some aggregates of the DTAB/MR/Sep@fluoro POS nanorods [40]. The reactions occurring in the formation of DTAB/MR/Sep@fluoro POS were analyzed by FTIR spectroscopy (Figure 8b). Compared with Sep and DTAB/MR/Sep, the new absorption bands at 1239 cm^−1^ and 1209 cm^−1^ corresponded to the stretching vibration of the C–F group, and the other band at 1151 cm^−1^ was assigned to silsesquioxane bands stemming from fluoro POS (hydrolytic condensation of PFDTES and TEOS) [40]. The surface chemical composition of the DTAB/MR/Sep@fluoro POS coatings was also studied by XPS spectra (Figure 8c,d). After treatment, the two fitting peaks at 291.7 and 294.1 eV were assigned to –CF_2_ and –CF_3_ of the Si(CH_2_)_2_(CF_2_)_7_CF_3_ groups due to the successful condensation of PFDTES and TEOS on the surface of DTAB/MR/Sep [41]. Moreover, the F1s peak at 688.8 eV was quite strong, and the F content was about 20.48 at% in DTAB/MR/Sep@fluoro POS, which was in accord with the above discussion. In summary, DTAB/MR/Sep hybrid nanopigments were successfully modified with fluoro POS.

To evaluate the superamphiphobicity of the DTAB/MR/Sep@fluoro POS coatings, the CAs and SAs of liquids are recorded in Appendix A and Figure 9a. Generally, water and glycerol had high CAs (>150°) and low SAs (<5°) on the coating, proving the excellent superamphiphobicity of the coating. In addition, the thermal and chemical stability of the superamphiphobic DTAB/MR/Sep@fluoro POS coating was investigated by a series of experiments under different conditions. DTAB/MR/Sep@fluoro POS exhibited outstanding stability at higher temperatures (180 °C for 0.5 h) since heat treatments had no effect on the CAs and SAs of water compared with that of fresh water. From Appendix A, DTAB/MR/Sep@fluoro POS was also resistant toward weakly acidic (0.1 M HCl) and basic solutions (0.1 M NaOH). The air cushion on the liquid–solid interface markedly enhanced the stability of DTAB/MR/Sep@fluoro POS toward these corrosive aqueous liquids. The chemical stability of the DTAB/MR/Sep@fluoro POS coating was then treated with 98% H_2_SO_4_ and 60% NaOH. As displayed in Figure 9b, the DTAB/MR/Sep coating was immediately wetted and lost its red color in an extremely acidic and basic medium once droplets of 98% H_2_SO_4_ and 60% NaOH came into contact with the coatings, which was consistent with previous reports [42]. Different from DTAB/MR/Sep, spherical droplets of 98% H_2_SO_4_ and 60% NaOH appeared on the DTAB/MR/Sep@fluoro POS coatings.

To investigate the color response of hybrid nanopigments and corresponding coatings under different atmospheres, the DTAB/MR/Sep powder and DTAB/MR/Sep@fluoro POS coating were put in an HCl or NH_3_ gas-filled desiccator, respectively. The samples with a red color were yellow in the acidic gas (Figure 9c). It was interesting to see that the color turned red again after the yellow samples were exposed to an alkaline gas. Furthermore, the reversible allochroic behavior between yellow and red could still be achieved after two acid/base cycles. The color changes of the DTAB/MR/Sep@fluoro POS coating were similar to those of hybrid nanopigments in different atmospheres. No influence on the CAs was observed after two acid/base cycles (Appendix A). The reversible acid/base allochroic phenomenon of the as-prepared samples was mainly dependent on the reduction and amination of the orange pigments with HCl and NH_3_ to obtain yellow pigments and red pigments, respectively [3,4,43]. Based on the rapid and highly reversible color switching of DTAB/MR/Sep hybrid nanopigments and the corresponding superamphiphobicity coating between different atmospheres, it is possible to prepare an intelligent coating with excellent stability for environmental monitoring.

## 4. Conclusions

In summary, the multifunctional intelligent hybrid nanopigments were fabricated combining Sep, MR, and cationic surfactants of DTAB and CTAB, in which the incorporation of surfactants effectively improved the adsorption capacity of Sep toward MR. The performances of the well-prepared samples could be greatly improved by the interaction among Sep, MR, and surfactants. Due to the high loading of MR after being modified with surfactants, the scavenging activity on DPPH and hydroxyl free radicals as well as the reducing power of the hybrid nanopigments with surfactants were higher than those of the unmodified ones. Furthermore, hybrid pigments significantly inhibited the growth of Gram-positive bacteria (*S. aureus*, *MRSA,* and *E. faecalis* strains). In particular, the MIC values of CTAB/MR/Sep towards *S. aureus*, *MRSA*, and *E. faecalis* reached 0.10 mg/mL. Based on the acid/base allochroic behavior of MR and the microstructure of hybrid nanopigments, DTAB/MR/Sep was selected to fabricate the gas-sensitive reversible allochroic superamphiphobic coatings by hydrolytic condensation of PFDTES and TEOS adopting a simple spraying method. The superamphiphobic coating presented excellent stability against high temperatures and corrosion in extreme acidic or alkaline solutions. Beyond all doubt, the intelligent acid/base-responsive allochroic hybrid nanopigments present great potential in gas detection.

## Figures and Tables

**Figure 1 nanomaterials-13-01792-f001:**
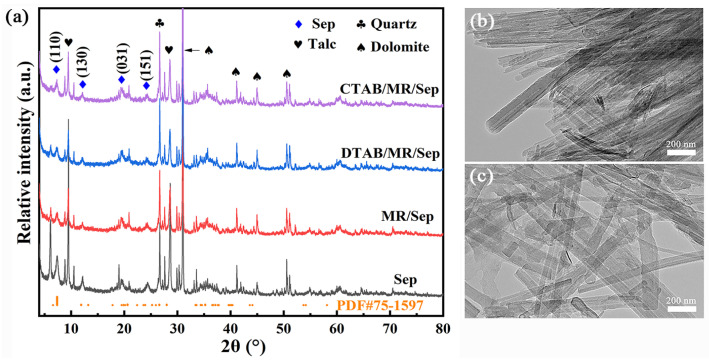
(**a**) XRD patterns of Sep, MR/Sep, DTAB/MR/Sep, and CTAB/MR/Sep. TEM images of (**b**) Sep and (**c**) MR/Sep.

**Figure 2 nanomaterials-13-01792-f002:**
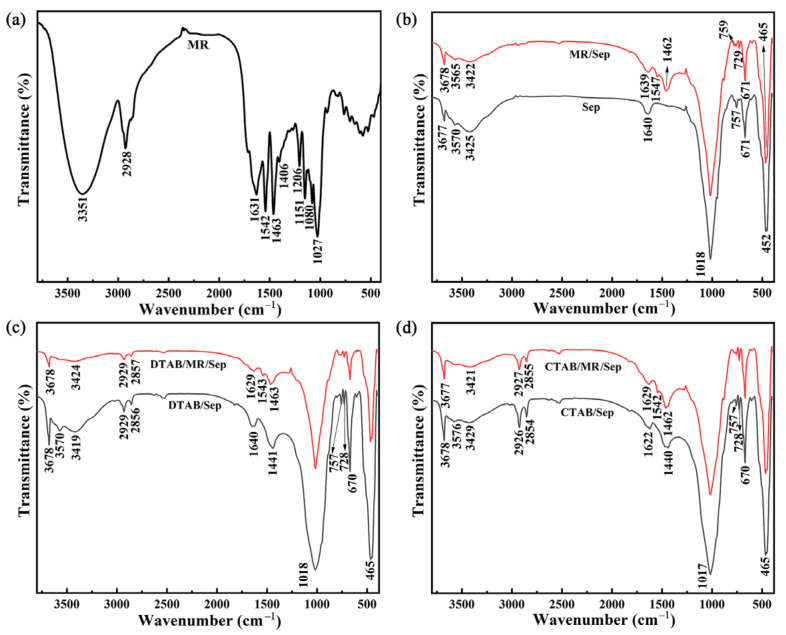
FTIR spectra of (**a**) MR, (**b**) Sep and MR/Sep, (**c**) DTAB/Sep and DTAB/MR/Sep, (**d**) CTAB/Sep and CTAB/MR/Sep.

**Figure 3 nanomaterials-13-01792-f003:**
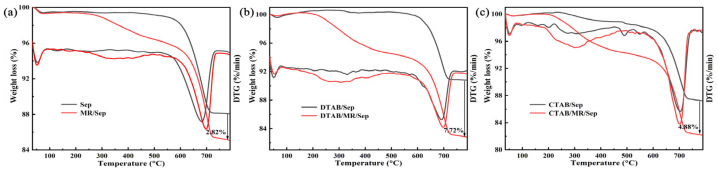
TGA and DTG curves of (**a**) Sep and MR/Sep, (**b**) DTAB/Sep and DTAB/MR/Sep, (**c**) CTAB/Sep and CTAB/MR/Sep.

**Figure 4 nanomaterials-13-01792-f004:**
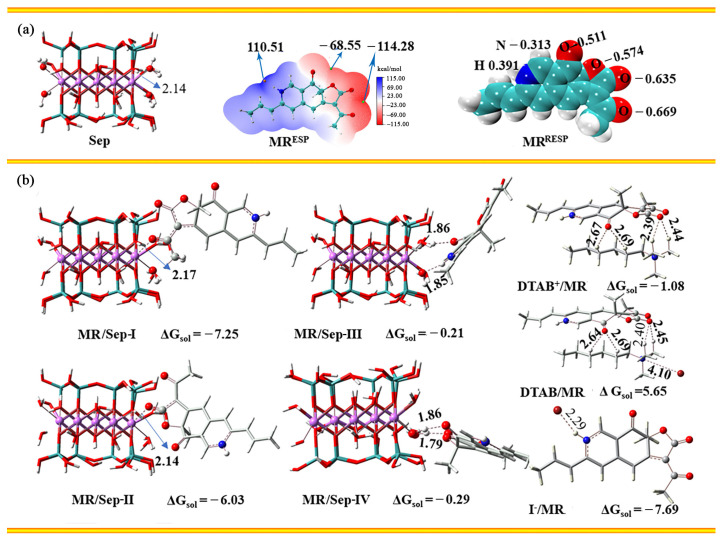
(**a**) The optimized model of Sep, the molecular surface electrostatic potential (ESP) and restrained electrostatic potential (RESP) atom charge of MR, and (**b**) The optimized adsorption structure between Sep and MR or DTAB and MR (the charge of Gibbs free energy is kcal/mol and the distances are in Å).

**Figure 5 nanomaterials-13-01792-f005:**
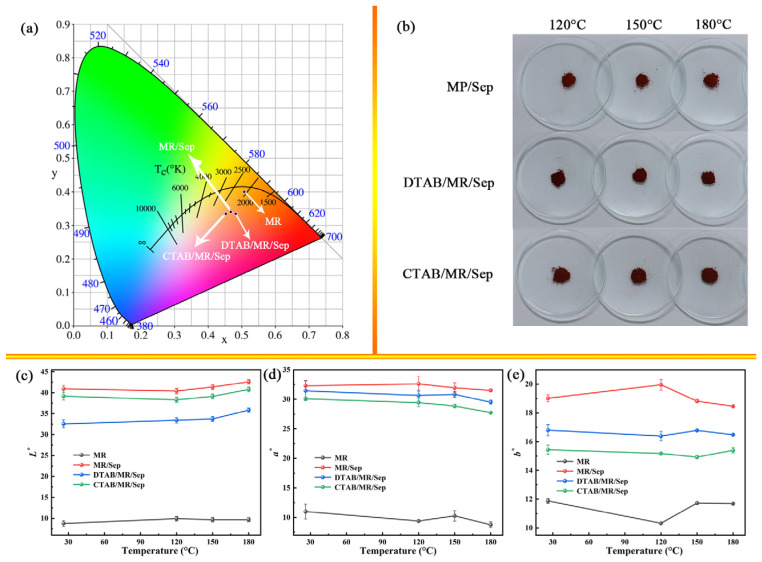
(**a**) Chromatic CIE coordinates of MR, MR/Sep, DTAB/MR/Sep, and CTAB/MR/Sep, (**b**) digital photos and (**c**–**e**) CIE of above samples after heat treatment.

**Figure 6 nanomaterials-13-01792-f006:**
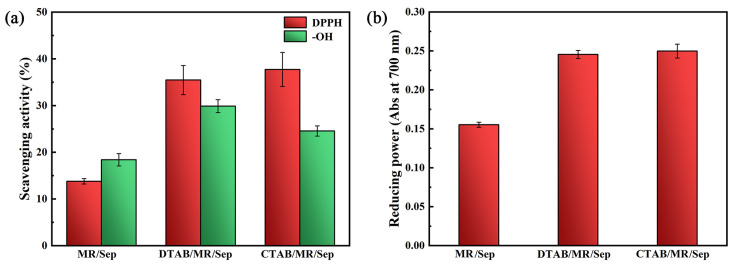
(**a**) Scavenging effects of MR/Sep, DTAB/MR/Sep, and CTAB/MR/Sep on DPPH radical and hydroxyl radical, and (**b**) reducing power (Abs at 700 nm).

**Figure 7 nanomaterials-13-01792-f007:**
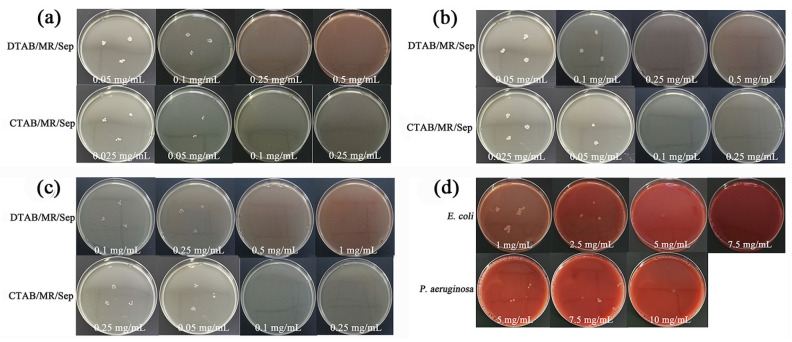
(**a**) *S. aureus*, (**b**) *MRSA,* and (**c**) *E. faecalis* treated with DTAB/MR/Sep and CTAB/MR/Sep at different concentrations, (**d**) *E. coli* and *P. aeruginosa* treated with DTAB/MR/Sep at different concentrations.

**Figure 8 nanomaterials-13-01792-f008:**
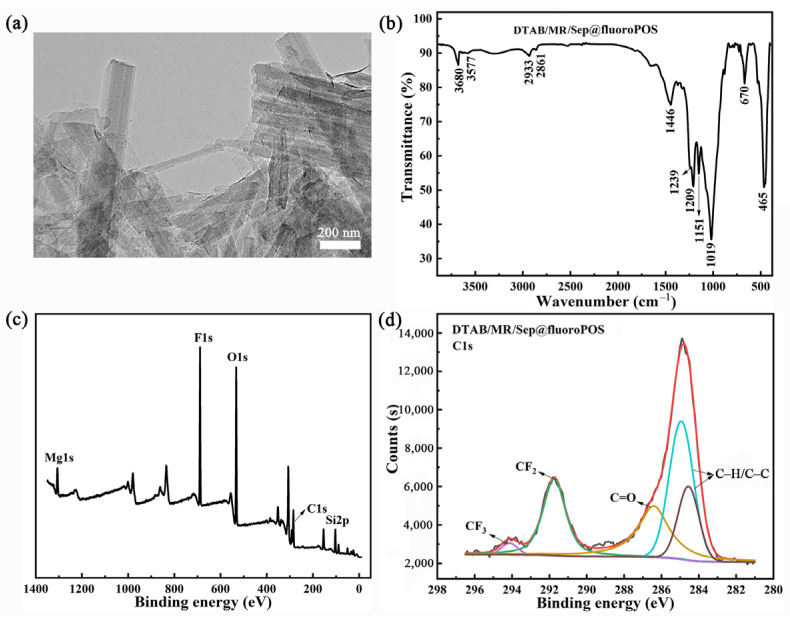
(**a**) TEM images, (**b**) FTIR spectra, (**c**) XPS survey spectra, and (**d**) high-resolution C1s spectra of DTAB/MR/Sep@fluoro POS coatings.

**Figure 9 nanomaterials-13-01792-f009:**
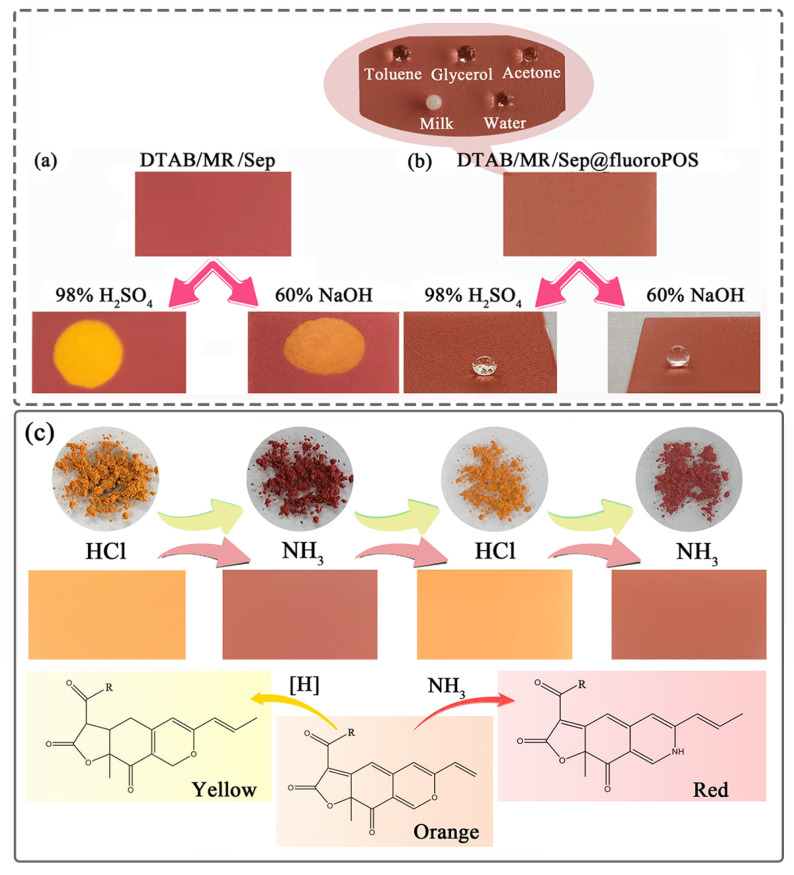
Digital images of the (**a**) DTAB/MR/Sep and (**b**) DTAB/MR/Sep@fluoro POS coatings before and after 98% H_2_SO_4_ and 60% NaOH corrosion, and DTAB/MR/Sep@fluoro POS coating with various liquids. (**c**) Digital images of the color change of DTAB/MR/Sep hybrid pigments and DTAB/MR/Sep@fluoro POS coating, as well as molecular structures under acidic and alkaline gas in turn (R = C_5_H_11_ or C_7_H_15_).

**Table 1 nanomaterials-13-01792-t001:** Pore structural parameters and zeta potentials of Sep, MP/Sep, DTAB/Sep, DTAB/MR/Sep, CTAB/Sep, and CTAB/MR/Sep.

Samples	*S_BET_* (m^2^/g)	*V_total_* (cm^3^/g)	Zeta Potentials (mV)
MR	-	-	−35.20 ± 1.63
Sep	31.09 ± 0.90	0.0516 ± 0.0011	−12.63 ± 0.46
MR/Sep	19.63 ± 1.63	0.0631 ± 0.0062	−16.53 ± 0.38
DTAB/Sep	30.57 ± 1.97	0.0760 ± 0.0014	−12.43 ± 0.31
DTAB/MR/Sep	13.27 ± 2.41	0.0389 ± 0.0100	−20.67 ± 0.21
CTAB/Sep	20.77 ± 5.31	0.0605 ± 0.0074	−2.05 ± 0.34
CTAB/MR/Sep	7.70 ± 1.33	0.0260 ± 0.0123	−19.57 ± 0.35

## Data Availability

The data presented in this study are contained within the article.

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
