# Peer review of "Preparation of Hybrid Nanopigments with Excellent Environmental Stability, Antibacterial and Antioxidant Properties Based on Monascus Red and Sepiolite by One-Step Grinding Process"

_nanomaterials, 2023, doi:10.3390/nano13111792_

Round 1

Reviewer 1 Report

Manuscript ID: nanomaterials-2414252

In the manuscript "Preparation of hybrid nanopigments with excellent environmental stability, antibacterial, and antioxidant properties based on Monascus red and sepiolite by one-step grinding process," the authors develop hybrid nanopigments with specific properties. These pigments are based on Monascus red and sepiolite.

The authors propose a one-step grinding process to produce these hybrid nanopigments. In this process, involves grinding Monascus red, sepiolite and cationic surfactants are milled together to obtain a homogeneous nanoscale mixture. This method aims to improve the stability and properties of the resulting pigments.

The proposed manuscript shows some interesting, although not definitive, results for understanding the potential applications of the materials. This point should be deepened to add value to the work.

The self-citations are numerous. They should be reduced.

Line 231 Page 6. The zeta potential values should be interpreted in more detail in the context of the different structure of the two cationic surfactants. It is not easy to understand why the value of -2.05 mV is attributed to addition of CTAB.

Figure 4 is not clear to understand the structural data.

Line 312 page 8. thermodynamic data should be explained in more detail.

English needs improvement.

Author Response

Question (1): The self-citations are numerous. They should be reduced.

Reply: Thanks for your comment. The self-cited references of No. 23, 26 and 43 were deleted from the manuscript according to the reviewer’s suggestion.

Question (2): Line 231 Page 6. The zeta potential values should be interpreted in more detail in the context of the different structure of the two cationic surfactants. It is not easy to understand why the value of -2.05 mV is attributed to addition of CTAB.

Reply: Thanks for your comment. Generally speaking, partial Si(IV) in Si-O tetrahedron and Mg(II) in Mg-O octahedron of Sep can be replaced by Al(III), Fe(III), etc. due to the isomorphous substitution, and thus Sep is negatively charged (-12.63 mV). As the typical cationic surfactants, dodecyltrimethylammonium cation (DTA+) or cetyltrimethylammonium cation (CTA+) has positive charge, and thus the formation of the electrostatic interaction between the surfactants and Sep resulted in the zeta potentials of DTAB/Sep (-12.43) and CTAB/Sep (-2.05) increased slightly, and the relevant statements have been revised in the manuscript. 

Question (3): Figure 4 is not clear to understand the structural data.

Reply: Thanks for your comment, the analysis of structural data is reasonable according to the knowledge and understanding of computational simulation, and some statements have been clarified in the revised manuscript.

Question (4): Line 312 page 8. thermodynamic data should be explained in more detail.

Reply: Thanks for your comment, and the relevant statements have been revised in the manuscript.

Reviewer 2 Report

The work carried out is of excellent quality and has called on multiple experimental approaches. But the study of the initial crystalline phases remained incomplete. The localization of the organic components is not convincing. I suggest a careful revision of the characterization section.

1)      The definition of initial components could have been detailed on a structural point of view. The chemical compositions of sepiolite and monascus red could have been given in the beginning of the text.

2)      The general formula of sepiolite retained in the literature is Mg4Si6O15(OH)2 · 6 H2O. This mineral belongs to the orthorhombic space group Pnan, with lattice parameters: a = 13.43 Å, b = 26.88 Å, c = 5.281 Å. In this work, the parameters are not given. The sepiolite XRD diffractogram of Fig. 1 is not identified with (hkl) indices. It is probably a mixture of phases (talc, dolomite….). A study of diffraction profiles could have delivered evaluation of average crystallite sizes of the various phases.

3)      A complete identification of the initial phases is required. The presence of significant amounts of dolomite CaMg(CO3)2 and talc Mg3Si4O10(OH)2 (in more or less purities) is observed in Fig. 1, in all crystallized systems discussed in this study. So, it is difficult to avoid any discussion about these two additional phases and on their roles.

Major revision is required.

No comments. 

Author Response

Question (1): A careful revision of the characterization section. The definition of initial components could have been detailed on a structural point of view. The chemical compositions of sepiolite and monascus red could have been given in the beginning of the text.

Reply: Thanks for your comment, the characterization section is described in detail, the main chemical compositions of Sep consisted of CaO (7.87%), Al2O3 (4.76%), MgO (25.66%), SiO2 (49.35%), K2O (0.93%), Na2O (4.93%) and Fe2O3 (1.39%) determined by X-Ray fluorescence (XRF) have been provided in the manuscript according to the reviewer’s suggestion. Furthermore, it has been confirmed that the chemical compositions of monascus red were composed of rubropunctamine and monascorubramine, their chemical structures were very similar except for the different side chain lengths (R = C5H11 or C7H15).

Question (2): The general formula of sepiolite retained in the literature is Mg4Si6O15(OH)2·6H2O. This mineral belongs to the orthorhombic space group Pnan, with lattice parameters: a = 13.43 Å, b = 26.88 Å, c = 5.281 Å. In this work, the parameters are not given. The sepiolite XRD diffractogram of Fig. 1 is not identified with (hkl) indices. It is probably a mixture of phases (talc, dolomite….). A study of diffraction profiles could have delivered evaluation of average crystallite sizes of the various phases.

Reply: Thanks for your comment, the general formula of sepiolite is Mg8(OH)4Si12O30(H2O)12, the lattice parameters was: a = 13.40 Å, b = 26.80 Å, c = 5.28 Å, and diffraction profile has been provided according to the reviewer’s suggestion.

Question (3): A complete identification of the initial phases is required. The presence of significant amounts of dolomite CaMg(CO3)2 and talc Mg3Si4O10(OH)2 (in more or less purities) is observed in Fig. 1, in all crystallized systems discussed in this study. So, it is difficult to avoid any discussion about these two additional phases and on their roles.

Reply: Thanks for your comment. Before use, Sep was treated with 4% hydrochloric acid to remove the associated impurity, especially dolomite. However, the diffraction peaks of dolomite and talc were still observed in Fig. 1, while the relative intensity was obviously decreased after acid treatment due to the decrease in the content of them. According to the chemical structure and physicochemical properties of monascus red pigment, it is difficult to load monascus red with the associated minerals. In fact, the unique pore structure of sepiolite is involved in the formation of hybrid materials, and the combination of monascus red pigment and sepiolite through electrostatic and hydrogen bonding is the key to stabilize monascus red pigment.

Round 2

Reviewer 2 Report

The manuscript was revised in the sense recommanded by reviewer. 

Accepted